# Safety Culture among Transport Companies in Ethiopia: Are They Ready for Emerging Fleet Technologies?

**Ehitayhu Hagos** [1], **Tom Brijs** [1], **Kris Brijs** [1], **Geert Wets** [1] and **Bikila Teklu** [2,*]

[1]   Transportation Research Institute (IMOB), UHasselt, Agoralaan, 3590 Diepenbeek, Belgium
[2]   Addis Ababa Institute of Technology, Addis Ababa University, Addis Ababa 1000, Ethiopia
*   Correspondence: bikilatek@yahoo.com

**Abstract:** The safety culture and safety climate of transport companies have a significant impact on fleet safety outcomes. Ample research shows that transport companies with a strong safety culture also show lower crash statistics. In spite of modern technologies that help with having a safer fleet, it is difficult to achieve a safer fleet without a proactive safety culture and climate. In Ethiopia, it is assumed that most transport companies have failed to create a distinguishable safety climate in their fleet safety administration and that their heavy vehicle drivers have a poor safety culture. These could be important factors contributing to a higher rate of road traffic crashes involving heavy vehicles. This study aims to assess the existing safety culture among a sample of transport companies in Ethiopia and identify suitable intervention methods to improve the safety culture. Moreover, the study sought to identify the readiness of the transport companies to apply modern technology in their fleets by examining their safety culture and safety climate. In total, 10 fleet managers and 174 heavy vehicle drivers participated in the fleet safety audit survey. A descriptive analysis and a detailed fleet safety audit score were calculated. Based on the scale scoring, ten companies score below best practices, one scores well below best practices, and only one company meets the criteria to be considered achieving best practices. The results from this study show that the safety culture and safety climate in most transport companies are quite limited. In addition, most transport companies implement similar safety measures, including inconsistent driver training and annual maintenance.

**Keywords:** safety culture; safety climate; transport companies; heavy vehicle drivers; fleet managers; fleet management; fleet technologies

## 1. Introduction

### 1.1. Driver Training and Rewarding System

Driver education and training are key components of a fleet safety management strategy. Various studies suggest that safety education and training should consist of an introduction to fleet safety policies, technologies, hours of service regulations, defensive driving, vehicle inspections, and behind-the-wheel training or mentoring [1,2]. To maintain safety, continuous education and training of fleet drivers is essential [3]. Accordingly, having well-trained drivers results in lower rates of road traffic crashes. Taking this into consideration, employers should carefully review an applicant's driving record regarding prior involvement in crashes or traffic rule violations. It is also recommended that, before hiring an applicant, employers should closely examine the driver's driving experience and training as well as recommendations from previous employer(s) [1]. Another strategy to strengthen fleet safety is a reward system of competitive compensation rates and performance-based incentives [1].

### 1.2. Emerging Fleet Technologies

There has been a revolution in vehicle safety technology in recent years. During the past 25 years, new vehicles have been expected to provide a high level of secondary

safety; however, more recently, technologies that tackle primary safety by focusing on crash causations and mechanisms are emerging [4].

On an international scale, we are witnessing a growing number of advanced technologies being implemented in fleets to improve the safety of drivers. Some of these technologies are speed warnings, collision warnings, headway monitoring, and fatigue detection. These fleet technologies are effective in reducing crash risk factors. For instance, technologies for detecting and countering task-related (TR) fatigue (caused by mental overload or underload) are proving to be effective tools for improving transportation safety [5].

Furthermore, having a well-established good safety climate is helpful for the fleet technologies' success. Fatigue technologies (i.e., on-board safety monitoring and lane departure warning) that were compounded with a good safety climate practice (revision of hiring policies, buddy project to pair new drivers with a trainer, improved safety pledge, and driver scorecards) achieved a 7% reduction in reported crashes, together with a 70% improvement in the CSA/BASIC crash indicator and a 49% improvement in the CSA/BASIC unsafe driving indicator [6].

### 1.3. Safety Culture

The term safety culture, in general, refers to the individual and group values, attitudes, perceptions, and competencies regarding safety [7]. Organizational culture is defined as the beliefs and values that pertain specifically to health and safety [8,9].

Safety culture in a fleet can be related to investing in well-examined safety technologies and programs, hiring the safest drivers, setting up compensation programs, creating a safety reporting system, encouraging driver health and wellness, and developing self-accountability for the safety of the fleet [10]. Fleets with a strong safety culture have a high commitment and dedication to safety from the fleet's executive team. Studies have shown that implementing a compensation program and programs that help improve driver health, incentives, and reward programs can be effective in improving driver safety [1]. Accordingly, research has shown that more safety-focused organizations report fewer crashes. In contrast, less developed safety climates can be associated with more dangerous behavior and a higher rate of crashes involving trucks [11].

In a commercial driving environment, a company's safety culture can be described through fleet managers' and fleet drivers' attitudes, values, and beliefs related to the safety and risk [12]. Varied research shows a direct relationship between safety culture and safe behavior [13]. Therefore, understanding the factors that influence a good safety culture can have a positive impact on an employee's overall safety performance. In research related to occupational safety, there is an increased focus on understanding safety culture given its potential impact on reducing traffic injuries and fatalities [14]. Any safety management system should incorporate learning from experience at all levels of the organization and through different formal and informal processes [15].

### 1.4. Safety Climate

Safety climate is operationalized as workers' shared perceptions of the organizational prioritization of workplace safety over other competing business [16,17]. It reflects characteristics such as management commitment to safety and safety communication [18,19]. Serving as a situational cue for organizational safety norms, safety climate promotes workers' safety behaviors and performance outcomes [20]. A series of meta-analytic studies have shown that safety climate predicts not only safety compliance and participation behaviors but also safety outcomes such as a reduction in crashes and injuries [21].

A positive safety climate has been shown to be associated with safe driving behavior on the job [22]. For truck drivers, the vehicle cab is their workplace; thus, any distraction on the job should be a concern for management and can be influenced by organizational factors. Hence, characteristics associated with the safety climate can be viewed as pre-

dictive in nature and may be useful when examining and assessing safety management within organizations.

*1.5. Road Transport Situation in Ethiopia*

Heavy vehicle crashes represent a significant proportion of road traffic fatalities and severe injuries around the globe. However, especially in developing countries the situation is critical. For instance, in Ethiopia, according to the most recent statistics (2019), the proportion of fatal accidents involving trucks out of the total number of fatal accidents equals 27.6% [23], whereas in a developed country such as Belgium, this proportion equals only 10% [24]. Hence, in comparison, the proportion of fatal accidents involving trucks is almost three times higher in Ethiopia. In most cases, heavy vehicles are owned by private transport companies and drivers regularly drive above the speed limit to reach their destination on time. Studies show that extended driving hours and fixed deadlines may contribute negatively to risky behaviors and habits such as smoking, drinking, and using psychotropic drugs [25]. These risk factors and risky behavior can have a significant impact on driver health and workability, as well as work safety, increasing the risk of injuries and traffic accidents [26].

Another risk factor is that some drivers can obtain a driving license illegally. Even through the legal framework to obtain a truck driving license in Ethiopia, the training period can take some time; furthermore, eligibility for the license states that the applicant shall be above twenty-two years old with a medical certificate that shows the applicant is free from any physical disability or adverse health condition that could make him/her unfit for the proper operation of a motor vehicle. Not only does the training period take three to five months, but there is also a requirement that drivers should renew their licenses every four years. However, if the driver is above fifty-five years of age, the license shall be renewed every two years. At each renewal, the medical examination result shall be presented. However, in order to avoid the lengthy training period, some drivers may obtain their driver's licenses from fraudulent companies. The legal framework also does not require the driver to take training while updating their license. Hence, having drivers with no or limited driving training increases the rates of traffic fatalities in Ethiopia.

Most drivers that work in private transport companies tend to drive faster to reach the destination earlier and work long periods without an appropriate break. Driving more than 5 h per trip and speeding leads to serious fatigue implications such as sleeping while driving and lack of focus on the road [27]. Most roads, especially rural roads, are not maintained. In addition, the street marks and traffic signs are missing in most cases, and as is typical of developing countries, drivers share the roads with livestock, which are an ever-present threat to safety.

Most local transport companies have heavy vehicles that are old and use outdated technology. Companies that own modern vehicles that have safety technologies struggle to find drivers that know how to use the new vehicle safety technologies. All these factors related to human errors, vehicle, environment, and transport company safety rules play a crucial role in the higher rates of traffic fatalities and severe injuries that occur on Ethiopian roads.

*1.6. Study Objectives*

This study aims to explore safety culture among a sample of transport companies that operate in Ethiopia, focusing on identifying the fleet safety management elements that should be improved for sustaining a positive safety culture. This study tried to address the issue, with the following research questions providing a deeper insight into the safety culture of transport organizations in Ethiopia. The main questions formulated for this research are the following:

1. What are the known safety measures taken by fleet managers?
2. Which practices are used to manage fleet safety?

3.   What is the role of standard fleet safety management practice perception by fleet managers and fleet drivers in the safety culture of the organization?

## 2. Materials and Methods

### 2.1. Material

The fleet safety audit tool developed by Mitchell et al. [28] was used to conduct the baseline analysis of safety culture of transport companies. Using a fleet safety management audit interview with fleet managers and drivers about fleet safety management practices, the tool assesses the safety culture in an organization. The audit tool provides standardized criteria that can be used by organizations to compare or identify their fleet safety culture against best practices. The fleet safety audit tool is used to identify the level to which fleet safety is managed within the company using best-practice techniques. More specifically, the audit tool assesses the management of fleet safety against five core categories, including (1) management, systems, and processes; (2) monitoring and assessment; (3) employee recruitment, training, and education; (4) vehicle technology, selection, and maintenance; and (5) vehicle journeys. Each of the five categories contains between one and three sub-categories. More specifically, there are twelve subcategories in total. Each subcategory receives, in the evaluation process, a score between 0 (for lousy performance) and 3 (for best performance), leading to a total score with a maximum of 36, as the interpretation indicated in Table 1. The categories are used to measure the qualities of management practice, focusing on management elements that can be verified. An organization rates its performance against the best practice in each sub-category at one of 4 levels from performing high standards to poor safety standards. Consequently, the total score aggregated over the five core categories can indicate how the organization performs in relation to best-practice fleet safety management (see Table 1).

**Table 1.** Fleet safety audit score description.

| Category | Poor | Well Below Best Practice | Below Best Practice | Approaching Best Practice | Achieving Best Practice |
|---|---|---|---|---|---|
| Score | 0–7 | 8–14 | 15–21 | 22–28 | 29–36 |

### 2.2. Participants

The sample size was determined using the following equation from Abaza, O. A., Arafat, M., and Uddin, M. S. (2021) [29]:

$$n = Z^2 pq / E^2 \tag{1}$$

where

n = the sample size,
Z = a number based on the confidence level,
*p* and *q* = the variances of the population,
E = the maximum error of the estimation.

The confidence level is 93% (Z = 1.81), and the margin of error is 7%. The estimation for both *p* and *q* is 0.5. The sample size calculation determined that a minimum of 167 distinct truck drivers were needed for the survey. Accordingly, twenty-four transport companies received a request letter to participate in this study. These companies were found through a list provided by the Ethiopian Federal Transport Authority. The companies were selected based on the type of goods they transport, the fleet size, and the location they work in. Ten transport companies agreed to participate in the study, and one fleet manager represented each company. Hence, in total, a hundred and seventy-four heavy truck drivers and ten fleet safety managers participated. All participants received a request letter for willingness to participate in this study and signed an informed consent letter.

*2.3. Procedure*

The transport companies were selected at random and were requested by letter to participate in this study by the researcher. Furthermore, the researcher explained the objectives and scope of the study to each transport company manager. Accordingly, the managers who agreed to participate contacted their fleet managers. The fleet managers were responsible for inviting 20 drivers from their company. After the invitation process was finalized, all the fleet safety audit tool items were explained by the researcher to every participant, and copies of the questionnaire with the cover letter explaining the research were provided. The participants answered the questionnaire anonymously.

**3. Results**

The results section discusses the findings obtained from surveys conducted among fleet managers and professional drivers. First, we will describe the fleet manager survey results, and then, we will discuss the data from the fleet drivers.

*3.1. Fleet Manager Participants*

The fleet audit survey and interviews were conducted with ten fleet managers from ten different transport companies. All ten participants held positions as fleet managers at their transport companies. Most of them have more than five years of experience in that position. They had on average four years of experience in their current position at their current company. The participants came to their current roles with backgrounds in fleet management (60%), administration, and management (40%). On average, the respondents were 38 years old ranging from 32 to 54 years, and all of them were male.

3.1.1. Fleet

Fleet managers' answers to the background questionnaire were summarized to develop an understanding of the nature of their fleet, its use, and management.

Some of the companies transport different goods, including packed foods, raw materials, garment materials, and steel. The rest transport companies transport cement and oil. The oil company mostly operates in but is not limited to the Ethio-Djibouti corridor. The rest of the transport companies' trucks operate throughout Ethiopia.

The participating organizations came from a range of different industry sectors, as shown in Tables 2 and 3.

**Table 2.** Participating transport company characteristics.

| Organization Characteristics | *n* |
|:---:|:---:|
| 5 to 10 years | 4 |
| >10 years | 6 |
| **Type of goods** | |
| oil | 1 |
| different goods | 7 |
| cement | 2 |
| **Size of fleet** | |
| <100 | 4 |
| 100–200 | 2 |
| >200 | 4 |
| **Sector** | |
| Commercial | 10 |

**Table 3.** Participating companies according to vehicle type.

| Vehicle Type | Company Name | | | | | | | | | |
|---|---|---|---|---|---|---|---|---|---|---|
| | TC1 | TC2 | TC3 | TC4 | TC5 | TC6 | TC7 | TC8 | TC9 | TC10 |
| Oil truck (48,000 L) | | x | | | | | | | | |
| Heavy truck (40–50 t) | x | | x | | x | | | | x | x |
| Heavy truck (>50 t) | | | | x | | x | x | x | | |

All participants indicated that the vehicles were owned by their company, except for two organizations (TC2 and TC6). The smallest fleet owned by one company was 27 heavy vehicles, and the company with the largest fleet had 413 heavy vehicles. All participants responded that they have a single driver for each vehicle. The average number of kilometers driven per year per vehicle was about six hundred thousand. However, not all participants provided the annual kilometers driven by each driver per year due to a lack of information. Moreover, the fleet managers reported that the drivers work on most days of the week.

### 3.1.2. Fleet Safety

Fleet managers reported that the fleets under their management had been involved in road traffic crashes, even though they were reluctant to report the exact number of crashes recorded by their fleet within the past 12 months.

When the fleet managers were asked about who holds the primary responsibility, 90% of the respondents indicated that the fleet safety managers and logistic managers held the primary responsibility for fleet safety. The remaining 10% of respondents indicated that the fleet manager and the risk manager have primary responsibility for fleet safety at their organization.

Although an attempt was made to include insurance data of the companies to obtain their crash rates compared to other transport companies, the insurance companies' regulation does not allow sharing of such data. However, the insurance companies claimed that the frequency of trips directly relates to the number of crashes.

### 3.1.3. Fleet Manager Interviews

This section summarizes the fleet managers' interview results. The first and second paragraphs discuss fleet safety practices, including providing training using GPS technologies. Next, the impact assessment of implemented safety measures, factors that assist in managing the fleets, and barriers to implementing best practices will be discussed. Finally, the safety practices that increase crash risk are covered.

All fleet managers reported that they use company policies and requirements to keep their fleets safe. All fleet managers pointed out that providing frequent safety training for drivers is one of the best methods to maintain fleet safety. Hence, they implemented theoretical training in their current organization. Even though they train their drivers, the time interval between training programs is wide. Four of them stated that they provide safety training only once a year. Three of them stated that they provide safety training only after drivers are involved in crashes. The remaining three respondents stated that they provide training once a year for all their fleet drivers and that they also give additional training for drivers involved in a crash. Accordingly, before they let the drivers drive again, managers ensure that drivers have taken safety training. The training mainly focuses on how to manage fatigue, how to reduce fuel consumption, how to manage workloads, and how to drive safely in general. However, six of the companies stated that usually, the annual training program might be skipped due to various reasons, such as cost.

Four out of ten companies also reported that they use GPS technology to locate vehicles and to manage the workload balance for the drivers as well as to avoid long driving hours and fatigue. The remaining six companies stated that they do not use GPS or other new technologies to manage the fleet. Instead, they follow various traditional management

methods such as calculating average fuel consumption based on the known distance between origin and destination. In addition, they calculate how many hours the trip will take from start to finish within the specified timetable. The fleet managers claim that these methods help their drivers to avoid driving beyond eight hours per day. All companies reported that they apply annual vehicle inspections as a safety measure. Furthermore, all of them stated that if a truck has been involved in a crash, they inspect the vehicle and make sure it receives the necessary maintenance before they put it back in business.

In total, 60% of the respondents stated that they did not perform impact assessments for their safety measures. For that reason, they find it difficult to assess the success of the measures they have been taking. Furthermore, fleet managers did not update the safety measurements they previously applied, which include annual inspection, safety training, and vehicle maintenance. However, the rest of the respondents mentioned that implementing GPS technology in their vehicles leads to improved information concerning fuel usage, speed, and the location of their fleet.

The 40 percent of respondents that adopted GPS technology stated that they changed from the traditional way of calculating fuel usage because they believe the traditional way of calculating fuel was not effective in determining precise fuel usage or monitoring drivers in remote areas. Fleet managers revealed that the main factors that assisted them in managing fleet safety in their organization were an effective management system, management, and organizational commitment and cooperation (see Appendix A) [28].

Fleet managers stated that the main barriers to implementing practices to manage fleet safety in their organization were time pressure, negative perception of driver training by drivers, a lack of resources for fleet safety, and driver behavior (see Table A2).

In total, 90% of the respondents believed that safety training and annual inspections reduce the risk of fleet crashes. In addition, 40% of the respondents replied that using various technologies such as a speed limiter and GPS helps them to reduce the risk of crash occurrences. Furthermore, 30% mentioned that in addition to training and using various technologies, having an agreement between the company and the driver helps fleet drivers to reduce the risk. Mostly, the agreement focuses on the consequences of being involved in a crash as a result of a driver's mistake. The consequence is typically a fine. Fleet managers were also asked to provide their opinion regarding the most essential factors to reduce the risk of fleet vehicle crashes. Accordingly, the most stated essential factor was driver training and education, followed by a culture of safety, an organization with a focus on safety, awareness, attitude towards fleet safety, and driver knowledge of the vehicle (see Table A3).

When asked whether any of the safety measurements in place were inadequate to their purpose, most responses were negative. Furthermore, respondents stated that there were no increases in the risk of vehicle crashes by keeping to the safety measurements that they had been following. However, three of the study participants replied that they thought that installing a large amount of vehicle technology could lead to a greater risk of fleet vehicle crashes (e.g., due to distraction).

### 3.1.4. Fleet Managers and Fleet Safety Management Factors

Based on their work experience, fleet managers were asked to describe whether specific factors had an impact on fleet safety management. All managers stated that in general, management systems and procedures have a positive impact on fleet safety management. Nevertheless, 40% of the respondents believed that a potential concern for the company's image (which is one of the components of the management system) does not have a significant impact. Having consultation between management and workers regarding safety was also rated as having an insignificant impact by 40% and 50% of respondents, respectively. All managers agreed that conducting driver performance monitoring and feedback has a positive impact on fleet safety management. Vehicle selection and maintenance were also rated by most fleet managers (90%) as having a significant positive impact on fleet safety.

All managers thought that employee recruitment, training, and education strategies were appropriate for managing fleet safety. However, half of the respondents thought the distribution of fleet safety newsletters would not be relevant to fleet safety. All managers agreed that performance-based incentives and disincentives, reviewing the route traveled by drivers for possible safety issues, using risk management strategies to reduce the risk of vehicle crashes (e.g., for speed, fatigue), a driver's attitude towards safe driving/ road safety, a driver's road traffic violation history (e.g., speeding tickets), and work pressure on drivers are having an impact in managing fleet safety. Some of the respondents (40%) stated that employing older or younger drivers had no significant impact on managing fleet safety.

### 3.2. Fleet Driver Participants

A total of 200 drivers spread across 10 transport companies provided informed consent to participate in the fleet safety audit written survey. However, due to reasons such as not having sufficient time off from work and a general lack of interest in the research itself, 26 of them dropped out of the study. Hence, in total, 174 drivers managed to participate in the study. Ninety-two percent of the drivers were professional drivers, and the remaining eight percent of the participants were assistant drivers. The number of years of driving experience was between 5 and 38 years. All participating drivers were male, and the average age of the drivers was 44.75 years.

#### 3.2.1. Fleet Driving

All the participants indicated that they drove a fleet vehicle on most days of the week, covering at least 100 km per day (see Table 4).

**Table 4.** Fleet use by participating drivers.

|  | **Number of Drivers** |
| --- | :---: |
| How often fleet vehicles are used | |
| On most days | 174 |
| Once or twice a week | |
| About once a fortnight | |
| Only occasionally | |
| Usual daily kilometers driven | |
| Less than 50 km | |
| 50 to 100 km | |
| 100 to 500 km | 174 |
| more than 500 km | |
| Vehicle allocation/ownership | |
| Dedicated for individual's use | |
| Pool vehicles | 174 |
| Company owned | |

Note: Dedicated for individual's use indicates vehicles that are allocated for employees' use only, not for business trips. Pool vehicles indicates vehicles that are owned by a company or organization for the use of its employees or members.

#### 3.2.2. Fleet Driver Safety

In total, 79% of the participating drivers reported that they had been involved in a crash in a fleet vehicle in the last 12 months. Most of these crashes (93%) resulted in property damage. None of the driver participants reported that they had been involved in an injury crash while driving a heavy fleet vehicle in the last year. When asked who was responsible for safety management at their company, all responded that the fleet manager held the primary responsibility in their organization.

### 3.2.3. Fleet Driver Interviews

In total, 82% of the respondents stated that safety training is one of the main safety practices in their organizations. This safety training mainly focuses on fatigue handling, speed management, and fuel management. Furthermore, 20% of the drivers stated that they were not aware of any kind of safety practice that has been promoted by their organization. All respondents stated that regular inspection is the primary safety practice in their organization. In total, 79% of the drivers are convinced that safety training is successful in managing fleet safety. However, the rest of the drivers responded that safety training is just a formality in their organization. According to 67% of the drivers, regular safety inspection of trucks is helpful in managing fleet safety, and 40% of the respondents indicated that new technologies such as GPS and speed limiters are also helpful in managing fleet safety. All the participants responded that they are not aware of any safety practice that has stopped due to ineffective results. Driver education and training, a reasonable period of driving per day (maximum 5 h driving without break), vehicle inspection, promoting safe driving, and a positive relationship with fleet managers were the main factors indicated by fleet drivers as essential in promoting fleet safety (see Table A4).

Fleet drivers were also asked about the major barriers when it comes to staying safe on the road, and most of them mentioned the poor condition of the road, poor road marking, interaction with corrupt police officers, driving for long hours per day, and reckless driving behavior of other drivers as the main contributing factors (see Table A5).

In total, 90% of the respondents believed that driver safety training, regular vehicle inspection, frequent maintenance of the vehicle, using in-vehicle technology, monitoring addiction to drugs and alcohol usage, a reasonable period of driving per day, and salary promotion for safe drivers are the main safety practices that have great potential for reducing the risk of fleet vehicle crashes and related injuries (see Table A6).

All drivers stated that they were unaware of any fleet safety management practices that had increased the risk of vehicle crashes. In total, 92% of them thought that the main risk factor in Ethiopia is speeding and, for that matter, safe speed training could help to manage fleet safety. Furthermore, 87% of the drivers also mentioned that road markings and bad road conditions are the main factors leading to a crash. Hence, improving the road condition or increasing drivers' awareness about the road condition that they will drive on would help to manage fleet safety. In total, 52% of the drivers stated that more could be done to manage fleet safety through regular driving license renewal. This would ensure that drivers update their driving skills and that they have all the necessary tests performed, such as eye testing. Adopting new technology features such as GPS and a speed limiter was stated by 78% of the respondent to be a good safety practice that could contribute to fleet safety. Fatigue management training was also stated by 67% of the respondents to be one of the areas where additional work could be done to increase the safety of fleets.

### 3.3. Fleet Safety Management Scores among Transport Companies

The interview output was analyzed to identify the transport companies' operational management performance. As explained in Section 2.1, the fleet safety audit tool is used for comparing companies' current performance to the best practices stipulated by the audit format [19].

The collected score for each fleet safety management component was added to calculate a total score. Based on the total score, one can identify how well the company's fleet safety management is performing. As shown in Table 5, most transport companies perform well in certain thematic areas, such as management commitment and fleet safety management. However, the cumulative result shows that the companies have fleet safety outcomes that are below best practices.

**Table 5.** Transport companies' operational fleet safety performance scores against best practices.

| Fleet Safety Practice Categories | Transport Companies (TC) with Their Fleet Safety Score | | | | | | | | | |
|---|---|---|---|---|---|---|---|---|---|---|
| | TC 1 | TC2 | TC3 | TC4 | TC5 | TC6 | TC7 | TC8 | TC9 | TC10 |
| **Management system and process** | | | | | | | | | | |
| Management commitment | 3 | 3 | 3 | 3 | 3 | 3 | 3 | 3 | 2 | 3 |
| Fleet safety management | 2 | 2 | 2 | 2 | 2 | 2 | 3 | 2 | 2 | 1 |
| Communication regarding road safety | 1 | 2 | 2 | 1 | 2 | 2 | 2 | 1 | 1 | 1 |
| **Monitoring and assessment** | | | | | | | | | | |
| Vehicle crash and incident investigation | 1 | 2 | 3 | 2 | 2 | 2 | 3 | 1 | 2 | 2 |
| Monitoring fleet safety performances | 0 | 1 | 1 | 1 | 1 | 1 | 2 | 1 | 1 | 2 |
| Performance monitoring and recognition | 0 | 1 | 0 | 0 | 0 | 1 | 2 | 0 | 1 | 1 |
| **Employee recruitment, training, and education** | | | | | | | | | | |
| Driver selection and assessment | 1 | 2 | 2 | 2 | 2 | 1 | 3 | 2 | 2 | 2 |
| Employee fleet safety induction | 0 | 0 | 0 | 0 | 0 | 0 | 2 | 0 | 0 | 0 |
| Driver training | 0 | 0 | 0 | 0 | 0 | 0 | 2 | 0 | 0 | 0 |
| **Vehicle technology, selection, and maintenance** | | | | | | | | | | |
| Fleet vehicle selection | 0 | 2 | 3 | 2 | 2 | 3 | 3 | 1 | 2 | 2 |
| Fleet vehicle maintenance | 2 | 3 | 3 | 3 | 3 | 3 | 3 | 3 | 2 | 3 |
| **Vehicle journeys** | | | | | | | | | | |
| Journey management | 1 | 1 | 2 | 1 | 1 | 2 | 3 | 1 | 2 | 2 |
| Total Score | 11 | 19 | 21 | 17 | 18 | 20 | 31 | 15 | 17 | 19 |

Note: Level I indicates the organization is performing at a high standard for this criterion and has a score value of 3. Level II indicates the organization is performing well for this criterion, but there is some room for improvement, and it has a score value of 2. Level III indicates the organization is performing reasonably on this criterion, but there is considerable room for improvement, and it has a score value of 1. Level IV indicates the organization is performing poorly on this criterion, with little to no activity, and has a score value of 0.

As demonstrated in Figure 1, even though the companies demonstrate different total score values, the majority (8 out of 10) of companies' performance is rated as "below best practice," except for two companies (TC 1: "well below best practice" and TC 7: "best practice"). This indicates that the fleet safety practice among different transport companies is more or less similar.

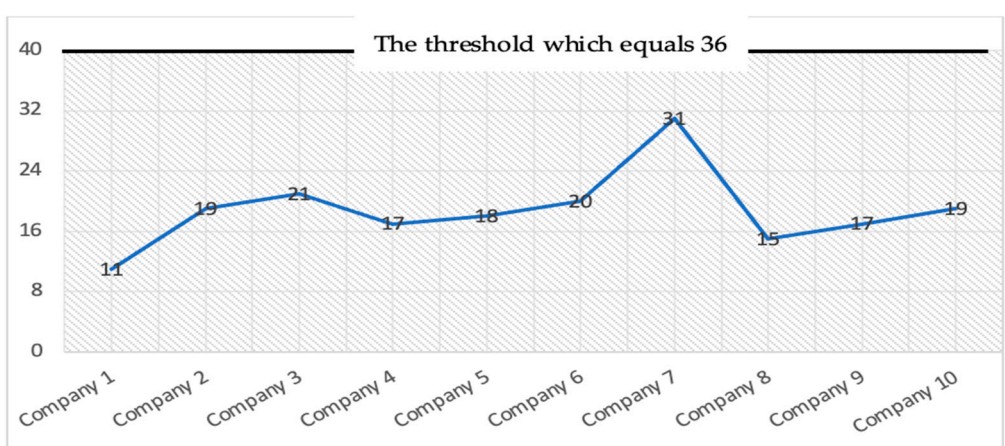

**Figure 1.** Transport companies' fleet safety management performance.

## 4. Discussion

This study investigated the existing safety culture in transport companies based on surveys conducted among heavy vehicle drivers and fleet safety managers. The results reveal several interesting findings. Despite minor management differences, all fleets used training and education, hiring criteria, and fuel usage control to improve their safety culture. Implementing this inclusive approach has been supported by previous studies [30–34]. However, fleet managers also mentioned barriers and challenges to implementing these strategies. For instance, 70% of the fleet managers stated that the training and education program is inconsistent due to a lack of budget. In addition, they reported drivers' resistance to training and education programs as a common barrier. In most cases, the drivers believe they are professional drivers and do not need additional training or education. The drivers stated that reasonable hours of driving, rewarding good driving behavior, salary promotion, and good communication with the fleet management team would improve safety outcomes (see Table 6). Drivers reported that poor road conditions, lack of road marking, reckless driving behavior of other drivers, corrupt police officers, and long driving hours per day are the main barriers to improving fleet safety. These driving behavior and road environment elements have also been mentioned in other research as causes for road crashes [35].

**Table 6.** Fleet manager and truck driver opinions on fleet management practices.

| According to Fleet Mangers | | According to Truck Drivers | |
|---|---|---|---|
| Practices that improve safety | Practices that are barriers to safety | Practices that improve safety | Practices that are barriers to safety |
| Standard hiring criteria | Inconsistent Training and education systems | Reasonable hours of driving | Poor road conditions |
| Training and education systems | Drivers' resistance to training and education programs | Salary promotion | Lack of road marking |
| Fuel usage control | | Good communication with the fleet management team | Reckless driving behavior of other drivers |
| | | Rewarding good driving behavior | Corrupt police officers |
| | | | Long driving hours per day |

Considering the results, the difference between the maximum and the minimum scorer is that the maximum scorer is a cement company with a bigger fleet size of 413 trucks and has a more developed safety culture. Additionally, the company has 11 years of work experience and more experienced international staff who work as fleet managers and assistant fleet managers. Moreover, beyond fuel management and workload, the company gives appropriate attention to other fleet safety elements, such as salary promotions for good drivers. On the other hand, the minimum scoring company has a smaller fleet size of 27 trucks with 5 years of experience and transports goods such as packed foods and different raw materials.

The perception of fleet safety by the company is quite limited. In fact, the manager focused mainly on fuel management and the time management of the drivers. In general, even though companies obtain different results for similar criteria, the difference in their safety culture is insignificant. This could be due to the similarity of regulation across different fleets. In most cases, the regulation does not give appropriate attention to various fleet safety elements, such as mandatory break times for periods of driving of more than four to five hours.

The results also highlight that all the participating fleets in the study used annual vehicle inspections as a safety measure. This supports previous research findings emphasizing the important role of vehicle maintenance and work environment usage in improving safety outcomes. Results also show that fleet drivers have a crash incidence average of

at least one accident per year. This could be due to the lack of important safety culture elements, such as a poor attitude towards learning or a lack of communication about safety. Varied research shows management commitment, communication, and journey planning as statistically significant predictors of safety outcomes [3]. Even though the fleet managers acknowledged the importance of elements of safety culture in improving fleet safety, they were not able to apply all elements to their fleet due to a lack of budget, lack of support from their management team, and driver resistance. These could be the reasons that most of the fleets (90%) in our study have a poor safety culture when compared to the best practices of developed countries.

Vehicle technologies such as GPS and speed limiters played a crucial role in improving the participating fleets' safety performance. Four companies implemented GPS to follow up on fuel usage, speeding, and the specific location of their trucks. These helped them to achieve a better safety outcome compared to the rest of the fleets that use traditional fuel usage calculations. These results are supported by ample research findings regarding safety technologies' effectiveness in improving safety. For instance, the active intelligent speed assist (ISA) system could prevent at least 10% of all severe crashes (and a higher proportion of fatal crashes) in Europe [36].

To keep fleets safe, all stakeholders should work together on a shared safety strategy. However, this study discovered that drivers and fleet managers have different views regarding fleet safety strategies. To be specific, all fleet managers reported that the fleet would be safe if the drivers took their annual fleet safety training and underwent the annual vehicle inspection. Based on this response, it can be concluded that the fleet managers believe the fleet safety strategy should focus on drivers. On the contrary, truck drivers believed that the fleet will be safer with improved road conditions such as working traffic signs and visible road markings. They also reported that road safety would improve if there were regular driver training programs as well as salary promotions for safe driving. These varying perceptions seem to come down to neither side wanting to take full responsibility, but, for real and sustainable improvements in fleet and road safety, all parties need to be on the same page.

The problem of blaming others and not taking responsibility for anything that has permeated society has caused road traffic crashes to worsen and has complicated the solution. Although future research will be needed to examine the impact of this avoidance of responsibility on road traffic safety, this is the greatest barrier to having a safer fleet or a safer traffic system in general. It is also important to note that even if roads incorporate all road safety elements, such as road markings and traffic signs, and trucks are equipped with safety technology, it will not be possible to develop the safe traffic system needed unless drivers and fleet managers stop blaming others and begin taking responsibility.

## 5. Limitation

Transport companies willing to participate in this study acknowledge and work to improve their fleet safety, which may explain their willingness to participate. The limitation of this study is this potential selection bias. We acknowledge that ten transport companies do not provide a sufficient sampling frame for generalization. Thus, future research with a larger sample size is needed to validate the findings.

The study is also limited because we used self-reported data, which could have promoted socially desirable responses. However, the participants were assured that their answers were completely confidential, and the impact was likely to be low.

## 6. Conclusions

In conclusion, this study attempted to identify the existing safety culture in ten different transport companies operating in Ethiopia. The study revealed that most transport companies have a safety culture that can be substantially improved. Survey and fleet safety audit analysis showed that most transport companies used inconsistent driver training

and annual maintenance as the main elements of the safety culture in their fleet. Some companies include GPS and speed limiter technologies in their fleet safety management.

To summarize, this study revealed that driver training programs and annual maintenance are the primary safety measures practiced by most Ethiopian transport companies. Furthermore, certain companies use GPS and speed limiter technologies in their fleet for safety and fuel management purposes. In addition, the study findings showed that even though fleet managers are aware of updated fleet safety management measures, they may need more time to apply them to their fleet for ample reasons, mainly cost-related and administrative issues. The study also showed that certain drivers perceive some safety measures, such as annual driver training, as unnecessary because they believe they are professional drivers with long driving experience.

These results support previous studies finding that driver training and education, rewarding drivers for good driving behavior, and using vehicle technologies to reduce crashes are essential for improving safety culture.

Due to the poor safety culture of these transport companies, implementing advanced safety technology in a fleet may not be effective for achieving positive safety outcomes. Thus, transport companies need to improve their safety culture along with implementing safety technology. Ultimately, this maximizes the effectiveness of safety technologies.

These findings will be useful in developing efficient safety culture elements across transport companies in Ethiopia.

**Author Contributions:** Conceptualization, E.H., T.B. and B.T.; Methodology, E.H.; Data curation, E.H.; Writing—original draft, E.H.; Writing—review & editing, E.H., T.B., K.B. and G.W.; Supervision, T.B., K.B., G.W. and B.T. All authors have read and agreed to the published version of the manuscript.

**Funding:** This study was supported by the Special Research Fund (BOF) of Hasselt University (BOF20BL07).

**Institutional Review Board Statement:** This study has been reviewed by the UHasselt Social–Societal Ethics Committee (SSEC) and has been approved (Approval number: REC/SMEC/VRAI/201/108).

**Informed Consent Statement:** Informed consent was obtained from all subjects involved in the study.

**Data Availability Statement:** The anonymized data presented in this study are available on request from the corresponding author.

**Conflicts of Interest:** The authors declare no conflict of interest.

## Appendix A

**Table A1.** Fleet manager opinions of the main factors that assist fleet safety.

| Main Factors That Assist in Managing Fleet Safety | *n* |
|---|---|
| Management and organization commitment and cooperation | 8 |
| Crash investigation and reporting, having good data available | 5 |
| Introduction and use of safe policies and practices, i.e., an effective management system | 10 |
| Good communication across the organization, including staff feedback | 2 |
| Use of new vehicle technology (e.g., vehicle safety features) | 4 |
| Engagement of the workforce, i.e., employees value safety | 1 |
| Strong safety culture in the organization | 4 |
| Resources to implement fleet safety programs | 1 |
| Implementing practices, i.e., putting safety into practice | 6 |
| Conducting observations of drivers | 1 |
| Conducting a licensing program with company drivers | 1 |
| Driver education and accountability | 8 |

**Table A2.** Fleet manager opinions of the main barriers to managing fleet safety.

| Main Barriers to Implementing Practices to Manage Fleet Safety | *n* |
|---|---|
| Lack of resources for fleet safety | 8 |
| Poorly motivated staff and poor attitudes towards fleet safety | 5 |
| Lack of management commitment to fleet safety | 2 |
| Lack of engagement by staff, apathy, and a lack of accountability for driving behavior | 2 |
| Large size of company (e.g., problems of access) | 0 |
| Misconceptions regarding in-vehicle monitoring | 0 |
| Lack of driver training | 0 |
| Driver behavior | 8 |
| Time pressures | 10 |
| Corporate structure | 6 |
| Lack of systems to put into place and knowledge of where to receive information from | 3 |
| Fleet safety procedures unknown to drivers | 2 |
| Lack of understanding of the latest technology | 3 |
| Use of both company and personal vehicles for work purposes | 0 |
| Perception of driver training | 10 |

**Table A3.** Fleet manager views of the essential factors to reduce the risk of fleet vehicle crashes.

| Essential Factors to Reduce the Risk of Fleet Vehicle Crashes | *n* |
|---|---|
| Driver training and education | 10 |
| Crash investigating and reporting, including benchmarking | 4 |
| Culture of safety, an organization with a focus on safety | 10 |
| Awareness, engagement, and attitude towards fleet safety | 10 |
| Fatigue management practices | 9 |
| Driver knowledge of vehicle | 10 |
| Use of new vehicle technology | 5 |
| Appropriate resources for fleet safety | 7 |
| Good recruiting practices | 4 |
| Ability to identify and manage risk while driving | 3 |
| Executive and middle management leadership and commitment | 4 |
| Examination of vehicle usage (i.e., video conference instead?) | 0 |
| Adherence to safe driving guidelines | 6 |
| Accountability of drivers for their driving behavior | 5 |
| Ability to modify driving behaviors | 1 |
| 3 s gap rule when driving behind another vehicle | 1 |

**Table A4.** Fleet drivers' thoughts on the main factors that promote fleet safety in the company.

| Main Factors Used to Promote Fleet Safety in the Company | *n* |
|---|---|
| Driver instructions regarding safe driving, education, and training | 100 |
| Vehicle inspections and routine maintenance | 96 |
| Reasonable period of driving per day | 100 |
| Safety information to staff (e.g., updating drivers regarding changes) | 58 |
| Driver familiarization and orientation to vehicle | 80 |
| Driver skills and awareness | 92 |
| Reporting of near misses and crashes and crash investigations | 47 |
| Use of new vehicle technology (e.g., vehicle safety features) | 73 |
| Resources (e.g., funds to stay in hotels on long trips instead of driving) | 62 |
| Promoting safe drivers | 98 |
| Written safety policies and procedures (e.g., alcohol and drug policy) | 53 |
| Driver attitudes, behavior, and motivation | 67 |
| Positive relation with fleet managers | 98 |
| First aid knowledge | 48 |
| Imposing limits on driving time and encouragement of breaks | 100 |
| Safety pre-start checklist for vehicle use | 86 |

**Table A5.** Fleet drivers' thoughts on the main factors that promote fleet safety in the company.

| Main Barriers to Staying Safe on the Roadway | *n* |
|---|---|
| Other driver behaviors, attitudes, and driver errors (e.g., not obeying road rules) | 100 |
| Poor roadways (e.g., single carriageway, windy, road works) | 100 |
| Long-distance driving and the need for breaks | 96 |
| Corrupt police officers | 98 |
| Ageing vehicle fleet and the need for a well-maintained vehicle | 86 |
| Driver distractions | 92 |
| Driver motivation | 72 |
| Lack of crash avoidance knowledge (e.g., safe speeds) | 10 |
| Wildlife | 0 |
| Lack of management commitment for fleet safety | 52 |
| Poor road marking | 100 |
| Lack of knowledge regarding company policies | 0 |
| Weather conditions | 0 |

**Table A6.** Fleet driver opinions of the three essential practices to reduce the risk of fleet vehicle crashes.

| Essential Practices to Reduce the Risk of Fleet Vehicle Crashes | *n* |
|---|---|
| Use of a safe vehicle that is well maintained | 92 |
| Practical driver safety awareness training | 96 |
| Driver familiarization and orientation to vehicle | 86 |
| Driver awareness | 92 |
| Fatigue management practices when driving long distances | 98 |
| Driver attitude | 56 |
| Good organizational policies and procedures | 78 |
| Management commitment to fleet safety | 74 |
| Drivers obeying the road rules | 92 |
| Speed management practices | 98 |
| Use of new vehicle technology | 84 |
| Getting drivers to treat the fleet vehicle as their own vehicle | 58 |
| Information regarding safe driving | 88 |
| Enviro-driving practices | 84 |
| Use of safety barriers on the roadways | 96 |
| Decreased workload (i.e., need for time) | 96 |
| Drivers not affected by drugs (khat) | 98 |
| Use of licensed drivers | 100 |
| Investigation of vehicle crashes | 78 |
| Conducting driver assessments | 54 |
| Safety pre-start checks prior to vehicle use | 80 |

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
