# Peer review of "Safety Culture among Transport Companies in Ethiopia: Are They Ready for Emerging Fleet Technologies?"

_sustainability, doi:10.3390/su15043232_

Round 1

Reviewer 1 Report

1In the abstractthe research implications need to be concise to focus. Instead, the findings and experimental results should be more specifically described, including some analysis results.

2This paper lacks a detailed description of the Emerging Fleet Technology, and it is suggested to add some practical data on the emerging fleet technology in terms of crash reduction.

3The description of the Table 1 part should be more detailed.

4The descriptions of the Table 2 and Table 3 are too simple, please expand the detailed description. And what is the relationship between the data and characteristics.

5It contains too much information in “3.1.3 Fleet Manger Interviews”, please make this part be more structured and provide data support.

6The respondents have different views on fleet safety in “3.1.2”. What are the other respondents' answers? How does the result relate to the discussion in this paper?

7It is recommended to add updated references.

Reviewer 2 Report

Title of paper: Safety Culture Among Transport Companies in Ethiopia: Are they ready for emerging modern fleet technology?

Manuscript Number: sustainability-2114071 

Comments 

The topic is very timely, and the review is within the scope of this journal. Commercial trucking is a complex social-technical system but exposed to the interaction with technology and human factors. Maintaining safe and risk-free operations of this transportation system is crucial to taking advantage of advanced knowledge about human factors and technological support for a safe outcome. Therefore it is an excellent call to try and map safety culture and its status to work proactively and predictively in managing this transportation system. 

Beyond that, the authors must address several concerns before this paper can be considered for publication. In the main, it demonstrates good thinking but a slightly relaxed approach to language, respect for the readers' understanding of the industry, and rooting in comparable industry's globally accepted regulations for more comprehensive scientific output. Please consider specific comments below [and comments in the PDF of the initial submission]. 

Please check our comments in the attached PDF.

Reviewer 3 Report

The authors have done a very good job of putting together a survey-based paper. Your thoughts and ideas were clearly reflected in the paper, backed by conclusive findings to provide safer transportation systems. I really liked the smooth writing style of the paper that communicates the information as simple and clear as possible. Great job!

My comments are summarized below:

1- I couldn't understand how the survey results would answer the question of the paper's title (Are they ready for emerging modern fleet technology?). To me, answering this question is bigger than the scope of the survey. There are many different things that should be explored to answer this question in addition to the safety culture in transport companies! I would encourage the authors to add to their review of literature the following publication to educate the readers regarding what transportation agencies should really look for to know if they are ready for modern fleet technology or not:

Hadi, M., Iqbal, M. S., Wang, T., Xiao, Y., Arafat, M., & Afreen, S. (2019). Connected vehicle vehicle-to-infrastructure support of active traffic management.

2- I really respect the authors' acknowledgement in the study limitation that the survey sample size of 10 companies is small. However, the reader still need to know the total population size to be able to build on these results in future research. For example, how many total transport companies in Ethiopia? No worries if the authors don't know the answer or even if the sample is too small, however, I would recommend the authors to add a section regarding the sample size calculation. Below is a simple example that the authors can cite and follow:

Abaza, O. A., Arafat, M., & Uddin, M. S. (2021). Physical and economic impacts of studded tyre use on pavement structures in cold climates. Transportation safety and environment, 3(4), tdab022.

The section is located in the above paper on (Page 6 - section 3.2.1 - Equation no.1). It is generally accepted if the authors assume the proportion of the population (p=0.5) and since (q=1-p), it's also fine to assume q=0.5 Also, please state what was the confidence level. It's acceptable if the authors want to assume low confidence level and this will somehow justify the small sample size.

Overall, good job. The article is good to go considering the comments above.

Round 2

Reviewer 1 Report

The comments have been well addressed.

Reviewer 3 Report

The authors have addressed all my comments. Thank you and good luck!